# A Multidisciplinary Approach Reveals an Age-Dependent Expression of a Novel Bioactive Peptide, Already Involved in Neurodegeneration, in the Postnatal Rat Forebrain

**DOI:** 10.3390/brainsci8070132

**Published:** 2018-07-10

**Authors:** Giovanni Ferrati, Emanuele Brai, Skye Stuart, Celia Marino, Susan A. Greenfield

**Affiliations:** 1Neuro-Bio Ltd., Culham Science Centre, Building F5, Abingdon OX14 3DB, UK; ss14372.2014@my.bristol.ac.uk (S.S.); celia.marino@neuro-bio.com (C.M.); susan.greenfield@neuro-bio.com (S.A.G.); 2School of Physiology, Pharmacology and Neuroscience, Faculty of Biomedical Sciences, University of Bristol, Bristol BS8 1TD, UK; 3School of Life Sciences, University of Warwick, Coventry CV4 7AL, UK

**Keywords:** development, basal forebrain, AChE-derived peptide, nicotinic receptors, optical imaging, Alzheimer’s disease

## Abstract

The basal forebrain has received much attention due to its involvement in multiple cognitive functions, but little is known about the basic neuronal mechanisms underlying its development, nor those mediating its primary role in Alzheimer’s disease. We have previously suggested that a novel 14-mer peptide, ‘T14’, could play a pivotal role in Alzheimer’s disease, via reactivation of a developmental signaling pathway. In this study, we have characterized T14 in the context of post-natal rat brain development, using a combination of different techniques. Ex-vivo rat brain slices containing the basal forebrain, at different stages of development, were used to investigate large-scale neuronal network activity in real time with voltage-sensitive dye imaging. Subsequent Western blot analysis revealed the expression profile of endogenous T14, its target alpha7 nicotinic receptor and the familiar markers of Alzheimer’s: amyloid beta and phosphorylated Tau. Results indicated maximal neuronal activity at the earliest ages during development, reflected in a concomitant profile of T14 peptide levels and related proteins. In conclusion, these findings show that the peptide, already implicated in neurodegenerative events, has an age-dependent expression, suggesting a possible contribution to the physiological mechanisms underlying brain maturation.

## 1. Introduction

The basal forebrain (BF) is an ensemble of distinct subcortical nuclei containing several cell types that are heterogeneous with respect to transmitter content, neuronal morphology and efferent outputs collectively projecting to multiple brain areas, such as the cortical mantle, hippocampus and olfactory bulb [1]. Accordingly, this telencephalic complex has long been implicated in cortical activation, attention, motivation, memory and, most importantly, in neurodegenerative diseases [2] such as Alzheimer’s disease (AD), where impairment and atrophy of this region are associated with early pathophysiological events [3]. Since the BF is rich in cholinergic neurons, developmental studies most commonly monitor the expression of two related enzymes, choline acetyltransferase (ChAT), and acetylcholinesterase (AChE) [4,5]. However, AChE can also exert a non-hydrolytic function, independent of cholinergic transmission [6,7], and is present in all vulnerable neurons prone to neurodegeneration [8], irrespective of their particular transmitter system. This non-enzymatic bioactivity has been attributed to a 14-mer peptide derived from its C-terminus (AChE-peptide, ‘T14’) with, interestingly, a strong homology to amyloid beta (Aβ) [9,10,11]. As an isolated peptide, completely independent of AChE and cholinergic transmission, T14 can mediate, via its binding to an allosteric site on the alpha7-nicotinic receptor (α7-nAChR) [10], calcium (Ca^2+^) influx, which exerts trophic or toxic actions, according to its dose, availability and age of the brain [10,12,13]. Among the different AChE isoforms, its monomeric form (G1), from which the AChE-peptide will have been cleaved, shows high levels both in early development and AD [14]. Interestingly, previous findings from our group revealed that the AChE-peptide has a similar expression pattern to G1 [14], being elevated in the same conditions [15,16]. Furthermore, former optical imaging experiments demonstrated that endogenous levels of T14 corresponded to changes in the neuronal network evoked responses in rat brain slices containing the BF, between 14 and 35 postnatal (P) day-old rats (P14 and P35) [16].

Interestingly, we recently showed that the endogenous AChE-peptide is significantly increased within the primarily vulnerable neuronal population and cerebrospinal fluid of Alzheimer’s patients [15]. Moreover, the exogenous administration of a synthetic version of the AChE-peptide drives an influx of calcium with eventual, secondary production of Aβ and Tau phosphorylation [15,17] through its interaction with the α7-nAChR [10], ultimately instigating a familiar cascade of pathological events characterizing AD. Since it has been suggested that neurodegeneration is an aberrant from of development, with T14 as a pivotal signaling molecule [12], the goal of this study was to characterize its role in development more fully. We aimed to use diverse, yet complementary, techniques to explore the effect of the peptide on neuronal network activity and protein-protein interaction with an innovative approach, which combines optical imaging, electrophysiology and Western blot, all on the same tissue. More specifically, we evaluated the evoked response at the meso-scale level of ‘assemblies’—extensive groups of neurons transiently-synchronized over a sub-second time scale [18,19,20]—using voltage-sensitive dye imaging (VSDI) in ex-vivo rat brain slices containing the BF. We then compared the parallel changes in population activity with the level of the endogenous T14 peptide and associated proteins, namely α7-nAChR, Aβ and *p*-Tau, all implicated in developmental and neurodegenerative processes [21,22,23,24]. In addition, rat brain tissue was used for co-immunoprecipitation (Co-IP) experiments to explore whether the interaction between the complex T14/nicotinic receptor [25] is also modulated in an age-dependent manner.

Our results indicate an age-related decline in BF neuronal activity during development, reflected in a concomitant profile of AChE-peptide levels and related proteins, along with the interaction of the T14-nicotinic receptor complex.

Overall, these data support the hypothesis that mechanisms underlying neurodegeneration and development could be similar, with T14 as a potential signaling molecule.

## 2. Materials and Methods

### 2.1. Animals

In this study were used 44 male Wistar rats divided into four groups of different ages, including Postnatal Day 7 (P7), 14 (P14), 35 (P35) and 60 (P60). In particular, 20 rats (five animals per age) were used in VSDI experiments followed by WB analysis on 4 animals per age-group. The remaining 24 animals (*n* = 5 for P7, *n* = 5 for P14, *n* = 7 for P35, *n* = 7 for P60) were used to perform co-immunoprecipitation on whole brain lysate and subsequent Western blot. All animal procedures were approved by the Home Office U.K. (according to “Schedule 1” regulations) and conducted in compliance with the requirements of the U.K. Animals (Scientific Procedures) Act 1986.

### 2.2. Brain Slice Preparation

Animals were anaesthetised using 1.5 mL 100% *w*/*w* isoflurane (Henry Schein, Cat # 200-070, North Chicago, IL, USA) in a sealed chamber (plastic box 30 × 20 × 15 cm) containing cotton buds for roughly one minute (min). Effectiveness of anaesthesia was confirmed by checking the hind paw-withdrawal reflex. After decapitating the animal, the brain was removed and placed in ice cold solution bubbled with carbogen (95% O_2_, 5% CO_2_) and containing (in mmoL): 120 NaCl, 5 KCl, 20 NaHCO_3_, 2.4 CaCl_2_, 2 MgSO_4_, 1.2 KH_2_PO_4_, 10 glucose, 6.7 HEPES salt and 3.3 HEPES acid; pH: 7.1. Three consecutive coronal acute slices (400-µm thickness) including the basal forebrain (1.20 to 0.00 mm from Bregma) [26] were sectioned using a Vibratome (Leica VT1000S) [16,17,27]. These slices comprehended different BF structures: the medial septum (MS); the diagonal band of Broca (DBB), consisting of a vertical diagonal band (VDB) and a horizontal diagonal one (HDB); the substantia innominata (SI), including the nucleus basalis of Meynert (NBM). Successively, each slice was divided along the midline (Figure 1a,b) to obtain two hemisections per coronal plane (giving a total of 6 hemislices per animal), which were transferred to a bubbler pot with artificial cerebro-spinal fluid (ACSF) (‘recording’ ACSF in mmol: 124 NaCl, 3.7 KCl, 26 NaHCO_3_, 2 CaCl_2_, 1.3 MgSO_4_, 1.3 KH_2_PO_4_ and 10 glucose; pH: 7.1) and incubated at 34 °C for 20 min. After, the hemisections were kept at room temperature (RT) for 30 min in oxygenated (95% O_2_, 5% CO_2_) ACSF to recuperate before VSD staining (Figure 1c).

### 2.3. Optical Recording Method

After recovery, hemisections were left in the dark in a moist chamber filled with bubbling ACSF. The voltage-sensitive dye Di-4-ANEPPS (4% 0.2 mM styryl dye pyridinium 4-[2-[6-(dibutylamino)-2-aphthalenyl]-ethenyl]-1-(3-sulfopropyl)hydroxide; Sigma-Aldrich, D8064, Darmstadt, Germany) [28] was dissolved in ACSF, foetal bovine serum 48%, DMSO 3.5% and cremophore EL 0.4%. This dye has been demonstrated to have minimal pharmacological side effects or phototoxicity and a high signal-to-noise ratio, making it suitable for the current study [29]. Hemislices were incubated for 20 min with the dye, then transferred and kept in ACSF (at RT, 22 °C ± 1.5 °C) for 1 h to wash off the dye excess and favour the recovery phase. Hemisections were then gently placed in the recording bath, continuously perfused with oxygenated ACSF and warmed to 30 °C ± 1 °C with a temperature control system (TC-202A, Digitimer Research Instruments, Hertfordshire, UK). Slices were maintained in position with a home-made plastic grid before placing the stimulating electrode (Pt-Ir concentric bipolar FHC electrodes, Bowdoin, USA; outer pole diameter 200 μm, inner pole diameter 25 μm) and the recording electrode in the BF area between the HDB and VDB region (Figure 1d,m).

### 2.4. Field Recordings

A glass pipette pulled from borosilicate glass capillaries (GC150TF-10, 1.5-mm outer diameter, 1.17-mm inner diameter; Clark Electromedical Instruments, Kent, UK) filled with 2 M NaCl solution containing 2% Pontamine sky blue 5BX dye (BDH Chemicals Ltd., Poole, UK) with a puller (Model P-87, Sutter Instrument Company, Novato, CA, USA) was placed in the region of interest (ROI), approximately 150 µm far from the stimulated area (Figure 1d). Field potential signals were amplified 1000-times using an IR-283 Amplifier (Neuro Data Instruments Corp., Delaware, PA, USA) coupled to a Micro 1401 mk II acquisition system (CED Ltd., Cambridge, UK) and displayed in Signal software (CED Ltd., Cambridge, UK), then saved and subjected to off-line analysis.

### 2.5. Data Analysis

VSDI and electrophysiology recordings were carried out simultaneously and performed as previously described [16]. Briefly, each experimental session consisted of a 25 min-long perfusion epoch, composed of a 15-min recording period preceded by a 10-min slice acclimatisation to the new environment, with an inter-stimulus interval (ISI) of 28 s. Field potential records produced 32 data frames (800 ms in length each) per perfusion condition. Each electrical stimulation (a 100-µs pulse of 30 volts) elicited a large artefact of about 2 ms in duration, whereas the field excitatory post synaptic potential (fEPSP) population activity trace deflection peak occurred 3–4 ms after stimulus delivery. For each experiment, every 4 frames out of 32 were averaged together, giving rise to 8 mean frames per recording. The maximum absolute values, corresponding either to a positive or negative deflection, were calculated for every average trace between 4 and 7 ms after stimulation. 

For VSDI recordings, data were collected and analysed as formerly indicated [16]. In summary, VSDI data were recorded in 4 × 4 mm 2-dimensional images, equivalent to 100 × 100 pixels with each pixel being 40 × 40 µm, from which data were extracted. The duration of VSDI recordings was approximately 15 min long per condition with an ISI between stimulations equivalent to that used for electrophysiology (28 s). Data analysis was carried out using a toolbox implemented in MATLAB [31] and previously described [16]. The fixed region of interest geometry was post-hoc visually depicted on the slice in order to include the entirety of the evoked VSDI responses of the BF area containing the MS, VDB and HDB (Figure 1j,k).

VSDI data show results from the selected ROI, equivalent for all the slices collected, plotted as an arrow of activity over space and time (‘space-time’ maps, Figure 1i,j), as summed fluorescence fractional change indicated by the value calculated from the area under the curve between 0 and 300 ms after stimulation delivery (ƩΔF/F_0_), or as averaged summed activity, indicated by the average of summed long-latency fluorescence fractional change. The colour map used in our study to display spatiotemporal activity employs warm and cool colours representing depolarization and hyperpolarization, respectively.

### 2.6. Tissue Homogenisation

After VSDI experiments, the hemislices were immediately placed into 1.5-mL tubes containing lysis buffer (Figure 1e), prepared by mixing 1× phosphate-buffered saline (PBS) (Fisher, Cat # BP2438-4, Fair Lawn, NJ, USA) with phosphatase (Fisher, Cat # 1284-1650, USA) and protease (Roche complete PIC, 04693116001, Indianapolis, IN, USA) inhibitors and homogenized with microtube pestles. This step was performed on ice to avoid tissue degradation. Subsequently, the brain lysate was centrifuged at 1000× *g* for 5 min at 4 °C, and the supernatant was transferred into a new tube and stored at −80 °C until use.

### 2.7. Co-Immunoprecipitation

Whole rat brains were homogenized in lysis buffer (0.05% SDS, 1 mM EDTA, 150 mM NaCl, 10 mM NaH_2_PO_4_, 1% Triton-X-100, 0.5% salt deoxycholic acid), containing protease (Roche complete PIC, Cat # 04693116001, Indianapolis, IN, USA) and phosphatase inhibitors (Fisher, Cat # 1284-1650, USA). The samples were then sonicated for 10 s (Homogenizer Status x 120, Cat # 60404, Heidelberg, Germany) and subsequently centrifuged at 16,300× *g* for 30 at min 4 °C. Afterwards, 500 μg (1 μg/μL) of the lysis fraction were pre-cleared by adding normal rat serum (Invitrogen, Cat # 01-9601, Fair Lawn, NJ, USA) and incubated for 2 h at 4 °C with gentle agitation. Next, 50 μL of magnetic protein A/G beads (Fisher, Cat # 88802) were added to the lysate and incubated for 2 h at 4 °C with gentle agitation. The samples were then spun at 9000× *g* for 4 min at 4 °C, and the resulting supernatants were used for co-immunoprecipitation (Co-IP). Twenty five microliters of Anti-AChRα7-agarose beads (Santa Cruz Biotechnology, Inc., sc58607, Dallas, TX, USA) were added into the pre-clearing samples and left overnight at 4 °C with gentle agitation. The next day, the samples were spun at 9000× *g* for 4 min at 4 °C, and the bead pellet was gently washed three times using 400 μL of lysis buffer. Between washes, the beads were centrifuged at 9000× *g* for 4 min at 4 °C. The target protein was then eluted from the beads by adding 0.1 M glycine, pH 2.5, into each tube, which was then placed on the rotating mixer for 20 min at 4 °C. The samples were spun at 9000× *g* for 10 min at 4 °C, then the supernatants were transferred into new tubes and neutralized by adding 5 μL 1 M Tris-HCl, pH 8, and quickly vortexed and centrifuged. These samples were then used for Western blot analysis to detect the T14 signal.

### 2.8. Protein Measurement and Western Blot Analysis

The protein content of the supernatant was determined as previously described [17] using the Pierce 660-nm Protein Assay (Thermo Scientific, Cat # 22660, Rockford, IL, USA), following the manufacturer’s instructions. WB aliquots were prepared by mixing the supernatant (final concentration of 10 µg/µL) with loading buffer (Bio Rad, Cat # 161-0747, Hercules, CA, USA) and β-mercaptoethanol (Bio Rad, Cat # 161-0710, Hercules, CA, USA). Once ready, the samples were heated at 95 °C for 5 min and then stored at −80 °C until use. Proteins were separated on 4–20% precast polyacrylamide gels (Bio Rad, Cat # 456-1094, Hercules, CA, USA) and blotted on PVDF membranes (Immobilon-P, Sigma-Aldrich, Cat # P2938, Darmstadt, Germany) with the wet transfer procedure. After this step, membranes were stained to evaluate the transfer quality with Blot FastStain (G-Biosciences, Cat # 786-34, St Louis, MO, USA) following the manufacturer’s indications and imaged using a CCD camera (G-Box, Syngene, Cambridge, UK) in order to determine the total protein content, which was subsequently used as loading control for the statistical analysis, as previously described [32,33,34]. The blots were then destained with warm distilled water and incubated for one hour at RT with a 5% milk solution, obtained mixing Tris-buffered saline (TBS) with blotting-grade blocker (Bio Rad, Cat # 170-6404, Hercules, CA, USA). After the blocking step, the blots were washed 3 times for 5 min with TBS Tween 0.05% (TBST) and then probed with primary antibodies diluted in 1% blocking solution (TBST mixed with 1% blotting-grade blocker), over-night at 4 °C. The next day, the membranes were washed 5 times for 5 min with TBST and then incubated with secondary antibodies diluted in TBST for one hour at RT, with gentle agitation. After the incubation, the blots were rinsed 6 times for 5 min with TBST followed by a final wash with TBS for 10 min. The proteins were detected with chemiluminescent substrates (ECL) (Bio Rad, Cat # 170-5061, Hercules, CA, USA), through the CCD camera system. After imaging, the original blots were cropped and adjusted for brightness and contrast with Adobe Photoshop CC 2015.

The primary antibodies, all diluted at 1:1000 were: rabbit anti-T14 (Genosphere, Paris, France); rabbit anti-nicotinic acetylcholine receptor alpha7 (Abcam, ab10096, Cambridge, UK); rabbit anti-amyloid beta (Cell Signaling, Cat # 8243, Danvers, MA, USA); mouse anti-phosphorylated Tau (Thermo Fisher, MN1020, Cambridge UK). The secondary antibodies, both HRP conjugated, were goat anti-rabbit (Abcam, ab6721, Cambridge, UK), diluted at 1:5000, and goat anti-mouse (Sigma-Aldrich, A9309, Darmstadt, Germany), diluted at 1:2000.

### 2.9. Image Processing

The immunoblots representing the total protein staining (Appendix A) were analysed as the loading control (LC) as previously described [32,33,34]. The intensity of the protein signal was measured using ImageJ software (NIH, Bethesda, MD, USA) and clearly summarized in the following video [35]. Briefly, the optical density of the loading control or the investigated protein was determined drawing a rectangular box surrounding each lane, in the total protein blot, or the band of the protein of interest. Then, the optical density within each square was calculated. Subsequently, the values corresponding to the LC were averaged in order to determine the coefficient of variation (CV) among the lanes. CV with low values reflected a homogeneous loading of the samples. After, the obtained values for the LC and proteins were analysed with Excel 2013 (Microsoft, Redmond, WA, USA), by dividing the band intensity value of the investigated molecule by the value corresponding to the same lane in the loading control blot. This step is performed to standardize the samples inter-variability. The results obtained per each age were further analysed normalizing the P14, P35 and P60 values against the P7 group, considered the reference age. In Appendix A are provided the uncropped blots showing the proteins of interest represented in the main figures. In Figure 4e, the input lane has been cut from the same membrane and placed at the beginning of the blot for clarity. All the blots were homogeneously adjusted for contrast and brightness with Adobe Photoshop (PS software CC 2015, San Jose, CA, USA) to reduce the background noise. After normalization, the data were statistically processed and plotted using GraphPad Prism 6 (GraphPad Software Inc., San Diego, CA, USA).

### 2.10. Statistical Analysis

For WB analysis, protein levels were normalized to total protein content, as previously described [32], and determined using ImageJ software (NIH, Bethesda, MD, USA). Statistical analyses of VSDI and WB experiments were performed using GraphPad Prism 6 (v6.05; GraphPad Software Inc., La Jolla, CA, USA), and as all the data were tested for normality, we used one-way analysis of variance (ANOVA) followed by Bonferroni’s multiple comparisons post-hoc test, considering the P7 group as the reference age. Moreover, we carried out an additional post-hoc test, “trend analysis”, to evaluate the linear pattern of the investigated proteins across the four ages. Both post-hoc tests were performed a priori regardless of the ANOVA result, because we wanted to further explore any potential age-dependent expression change within the studied time window. For all statistical tests, *p* < 0.05 was considered significant; data were expressed as the mean ± SEM. Statistical significance: *^,#^
*p* < 0.05; ** *p* < 0.01; *** *p* < 0.001; ****^,####^
*p* < 0.0001.

## 3. Results

### 3.1. Basal Forebrain Population Activity Measured with Optical Imaging and Electrophysiology in Postnatal Development

VSDI was used to record meso-scale neuronal network responses to direct stimulation of the BF in ex-vivo rat brain sections. The VSDI signal, shown as a wave of activity recorded with a high spatial (micrometres) and temporal (milliseconds) resolution, was represented as a ‘space-time’ map (Figure 2a). A clear activation profile presenting an age-dependent reduction in peak activity and a lower response could be distinguished throughout developmental stages. More specifically, the quantification of activity in equal regions of interest (ROIs) revealed a decreasing depolarization pattern from strongest in P7 to weakest in P60 animals. The size of the activated area diminished accordingly with the maturation of the network, with a weaker signal intensity at the outset in older animals protracting over time (Figure 2a).

The area under the curves comprised between 0 and 300 ms after the stimulus delivery defines the summed fluorescence fractional change data (Figure 2b). The early postnatal period (P7–P14) was defined by a common profile, with a peak at 5 ms after stimulation onset preceding a continuous decline until approximately 40 ms (Figure 2b). A small rebound in response lasting respectively about 25 ms for P7 and 10 ms for P14 was detectable until another constant long-lasting reduction appeared (Figure 2b). Maximum responses for P35 and P60 were much smaller and slightly delayed in comparison to younger groups. At both ages, a 20-ms decrease phase preceded the rebound in activity followed by the long-lasting decay (Figure 2b).

Summed fluorescence values relative to BF long-latency activity (Figure 2c) indicated a highly significant difference across the four groups, illustrating a strong reduction of neuronal assembly magnitude with maturation (F_(3,120)_ = 9.613, *p* < 0.0001, one-way ANOVA). Comparative analyses between two datasets pinpointed a significant BF-evoked response change between P7 and all other groups (P7 vs. P14, *p* = 0.0280; P7 vs. P35, *p* < 0.0001; P7 vs. P60, *p* < 0.0001, Bonferroni’s test) (Figure 2c). In line with this observation, also the trend analysis across ages was strongly significant (*p* < 0.0001). 

Concomitant with VSDI, we recorded local field potentials (LFP) placing a recording electrode in an area surrounding the vertical diagonal band (VDB) at a distance inferior to 150 µm from the stimulating electrode (Figure 1d). LFP corresponded well with VSDI signals (Figure 3a, bottom), confirming that these techniques are complementary in reflecting large neuronal population dynamics in real time. Both signals revealed a clearly correlated activity, but while VSDI showed a slower rising phase, reaching a peak amplitude at around 6 ms, field excitatory postsynaptic potentials (fEPSPs) exhibited their maximum after approximately 3 ms (Figure 3a, bottom). The LFP signal decay (Figure 3a, bottom) was much faster compared to VSDI (see Figure 3a, top, for the complete trace).

The averaged maximum amplitudes indicated a difference across ages. At young stages, although most of the fEPSPs showed low response amplitudes, there was a significant number of recordings characterised by a much stronger signal magnitude, as evinced by the long-tail distribution in the graphs (Figure 3b). This effect varied as age increased and the connections matured, displaying a higher incidence of lower amplitude fEPSPs and a reduction (already visible at P35) or total absence (at P60) of greater responses (more than 80 µV) (Figure 3b).

### 3.2. Protein Levels Change Differentially during Postnatal Development

AChE-peptide expression was not different across groups (F_(3,20)_ = 2.222, *p* = 0.1171, one-way ANOVA) (Figure 4b). However, since the *p*-value was around the trend level, we additionally performed post-hoc analyses. While comparing P7 with the other age-groups, T14 was not changed (P7 vs. P14, *p* > 0.9999; P7 vs. P35, *p* = 0.5807; P7 vs. P60, *p* = 0.3107, Bonferroni’s test) (Figure 4b), the linear trend was significant in the explored postnatal period (*p* = 0.0397).

These results are also in line with previous evidence indicating that the endogenous monomers of the AChE-peptide, measured with ELISA on rat whole brain lysate, were continuously reduced in the same age-range investigated here [16].

The nicotinic receptor levels were significantly changed across ages (F_(3,20)_ = 48.90, *p* < 0.0001, one-way ANOVA) (Figure 4d). We identified a substantial diminution between P7 rats and older groups (P7 vs. P14, *p* < 0.0001; P7 vs. P35, *p* < 0.0001; P7 vs. P60, *p* < 0.0001, Bonferroni’s test) (Figure 4d). Similar evidence was detected when evaluating the trend analysis (*p* < 0.0001).

We observed after the Co-IP assay that the interaction between T14 and α7-nAChR showed, although not significant, a decreasing tendency among the four groups (F_(3,20)_ = 2.401, *p* = 0.0979, one-way ANOVA) (Figure 4f). When independently comparing the ages, we only detected a significant decline between P7 and P60 animals (P7 vs. P14, *p* = 0.5318; P7 vs. P35, *p* = 0.2018; P7 vs. P60, *p* = 0.0481, Bonferroni’s test) (Figure 4f); however, the linear trend analysis showed a marked profile (*p* = 0.0160).

Amyloid beta was significantly diminished across ages (F_(3,20)_ = 14.17, *p* < 0.0001, one-way ANOVA) (Figure 5b). We detected a strong age-related progressive decline from P7–P60 animals (P7 vs. P14, *p* = 0.0009; P7 vs. P35, *p* = 0.0050; P7 vs. P60, *p* < 0.0001, Bonferroni’s test) (Figure 5b). Consistent with this observation, the linear trend in the four groups was highly significant (*p* < 0.0001). Phosphorylated Tau analysis revealed a significant age-dependent variation amongst the four groups (F_(3,20)_ = 6.54, *p* = 0.0029, one-way ANOVA) (Figure 5d). In contrast, when comparing the P7 group against the others, we did not observe any change (P7 vs. P14, *p* = 0.2200; P7 vs. P35, *p* = 0.0626; P7 vs. P60, *p* = >0.9999, Bonferroni’s test) (Figure 5d). In accordance, also the trend analysis did not reveal any substantial change (*p* = 0.0706).

## 4. Discussion

### 4.1. Neurodegeneration as a Recapitulation of Development: A Common T14 Signalling System

Previous studies have long implicated the T14 peptide, cleaved from the C-terminus of AChE, in neurodegenerative events [9,14,15,16,17], as a recapitulation of its role in development.

During adulthood, the full AChE molecule most commonly consists of four catalytic subunits (G4), whereas during early developmental stages and in AD, it is found primarily as a monomer (G1) [8,14]. This pathological reversal back to a monomer has been attributed to an inability to oligomerize due to the absence of disulfide bonds containing the C-terminal peptide, which has been cleaved to play a role in degeneration, as it would have previously in development: hence, the dominance of G1 in development and AD is most importantly indicative of the cleaved T14 peptide, the key bioactive molecule in development and degeneration [12,36].

Given these findings, we have explored here whether T14 may contribute to postnatal brain development.

These results indicate an age-dependent reduction of network activity, which is paralleled by a decrease in the expression profile of T14, in particular when bound to its target alpha-7 receptor.

### 4.2. Optical Imaging of Age-Dependent Neuronal Activity

VSDI and electrophysiological data in the rat brain revealed a coherent profile of BF-evoked activity in the first two months, where neuronal assembly size decreased with age. Field recordings exhibited a pattern characterized by a more local, but less predictable response to stimulation during early postnatal stages, followed by a more spread and predictable behaviour as the neuronal wiring refines and undergoes maturation with aging, consistent with an effect of circuit reorganization [37]. Optical imaging and electrophysiology were combined to integrate different data relative to large-scale network dynamics and more confined neuronal interactions during BF development. This combining of techniques provides some advantages: first, although both techniques describe a population measurement, they likely have different integration areas, with LFP being more local than VSDI signals [38]. Second, VSDI recordings mainly report subthreshold changes in activity within the neuropil [39], whilst electrophysiology reflects somatic and axonal signals, thus providing complementary information. Third, in our experiments, we showed a strong correspondence in the evoked-response pattern between the different methods, confirming that the optical measurements are free from artefacts and indicate a real physiological effect. Moreover, field potentials also ensured that slices remained healthy throughout the whole duration of recordings [40], with viability confirmed by the stability over time of electrophysiological signals in response to stimulation at all ages.

On the other hand, with VSDI, it was possible to visualize changes in population activity over a larger spatial scale and longer time window than possible with extracellular electrophysiology. The reduction in both peak and long-latency VSDI signal observed with increasing postnatal age was not unexpected: assemblies, operating as networks of neurons coherently communicating to perform behaviourally-relevant tasks, are stronger during cortical development [18]. Therefore, the wide and robust activity detected in the BF during early postnatal stages (P7–P14) and decreasing with circuit maturation is in accordance with an age-dependent reduction in plasticity [16,18], where AChE-peptide has proven to play a modulatory role contributing to the change in VSDI signal. In support of a direct receptor interaction of the peptide on network activity, we have recently reported how a cyclized variant of T14 (NBP-14) blocks the interaction of its endogenous linear counterpart with the target receptor [16]. Moreover, it has already been shown that the AChE-peptide can initiate an intracellular cascade, instigating a feed-forward chain of trophic events resulting in neuronal differentiation, cell growth and cell adhesion [15]. Previous observations indicating a T14 reduction in whole brain lysates [16] across the same age-range used in this study are in line with our WB analyses, showing a decrease of its levels from the prepubertal phase (P14) to young adulthood (P60). This trend is in parallel with assembly magnitude, further supporting an involvement of the T14 in modulating network connectivity, through calcium-mediated mechanisms regulated by its interaction with α7-nAChR.

### 4.3. Parallel Developmental Changes in Levels of T14 and Related Key Proteins

Age-related responses in real time, where the basal forebrain was specifically targeted by optical imaging, paralleled trends in levels of the neurochemicals revealed by WB, which out of necessity involved evaluating the slice as a whole. Whilst we cannot attribute all the observed biochemical changes therefore to the basal forebrain exclusively, the close correspondence with neuronal activity in the basal forebrain suggests that this key region could be a determining factor of the effects observed. We detected several bands for T14, but the most consistent and prominent was at 50 KDa, suggesting that this peptide, as confirmed by Co-IP results, forms a complex with its target receptor α7-nAChR in brain tissue, as formerly described [25]. This observation is supported by previous studies demonstrating that its monomeric form of 2 KDa [11] can also interact with other proteins, like α7-nAChR, generating complexes of higher molecular weight [41].

The T14 pattern detected here shows a diminishing trend as previously observed on whole brain homogenate from age-matched animals, [16]. Moreover, this profile is in line with earlier findings describing how G1, T14 parental protein, also declines during postnatal development [14], as indicative of free bioactive peptide [12,15].

The age-related decrease of the nicotinic receptor is consistent with previous studies showing that, in rodent hippocampus and cortex, its levels are higher during early postnatal life, participating to synaptogenesis and dendrite branching phases, and downregulated in later stages, in concomitance with the progressive strengthening of the connections [21,42,43].

In addition, we observed that the physical interaction of T14 with its target receptor gradually declined in postnatal brain maturation. These data parallel the reduction in population activity measured with VSDI, therefore suggesting a possible contribution of this molecular complex to the modulation of the developing neuronal network.

The progressive amyloid beta decrease is likely associated with a diminished requirement during late phases of brain development. Although primarily recognized as a feature of neurodegeneration, several studies have demonstrated that low concentrations of Aβ contribute to neurogenesis, calcium balance and metal ion sequestration [44]. Moreover, given the sequence homology between T14 and Aβ and their decline from early postnatal to adult stage, it is interesting to speculate that T14, like Aβ, is not merely involved in neuropathological processes, but could also have a beneficial physiological function during development. This similarity between mechanisms of development and neurodegeneration had previously prompted the notion that the latter is an altered occurrence of the former [8,9,45].

The more homogeneous trend observed in p-Tau may underlie its importance during all postnatal development phases, being involved in microtubule (MTs) dynamics [46,47]. In addition, the presence of two bands around 60 and 70 KDa, detected in P14 rats and clearly separated at P35 and P60, is consistent with former studies indicating the cleavage of different isoforms with increased molecular weight related to aging [46,48].

## 5. Conclusions

In conclusion, taken together, the results of this study suggest that during BF development, endogenous T14 peptide shows a dynamic profile, whereby its availability falls and its interaction with the target α7-nAChR shows a disproportionate decline relative to the dwindling levels of receptor. This decrease in T14 at the nicotinic receptor can reduce the modulatory effect on calcium influx [15], resulting in the decline in activity [16] and protein expression [15,17] also observed here. The age-dependent reduction in T14 levels resulting in less calcium entry would in turn affect a cascade of intracellular events, including the modulation of voltage-sensitive calcium channels [49] and several protein kinases, such as GSK3 and MAPK, involved in the regulation of the Tau and amyloid pathways [15].

Overall, these results indicate that all the proteins studied here undergo an age-dependent change; however, the most sensitive parameter matching neuronal activity is the complex of the T14-nicotinic receptor, in turn suggesting a physiological role based on the interaction of these molecules during development.

Beyond these specific results, to the best of our knowledge, the combination of optical imaging, electrophysiology and Western blot in the same brain tissue represents a novel approach for gaining insights into general brain processes, such as in development. This strategy could be applied to explore large neuronal population and biochemical interactions of other brain structures in a site-specific and time-dependent manner.

## Figures and Tables

**Figure 1 brainsci-08-00132-f001:**
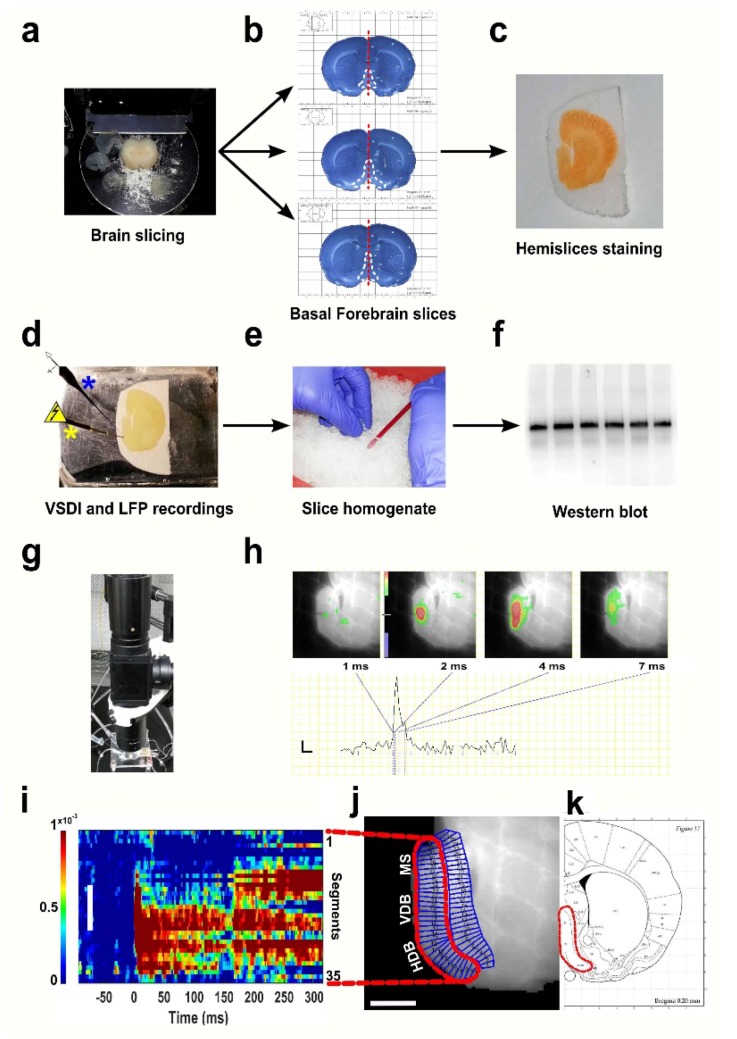
Schematic of the technical and experimental approach adopted in this study. Rat brains at four different ages (Postnatal Day 7 (P7), P14, P35 and P60) were sectioned (**a**) to obtain basal forebrain (BF)-containing slices [30] (white dashed lines) (**b**), which were subsequently cut along the midline (red dashed line in (**b**)) and stained with the dye molecule (**c**). After recovery, hemislices were transferred in the recording chamber (**d**) and set up with a stimulating electrode (yellow asterisk) accompanied by a recording capillary for electrophysiology measurements (blue asterisk). After voltage-sensitive dye imaging (VSDI) experiments, brain tissue was homogenised (**e**) and successively used for WB analyses (**f**). The VSDI setup employed for optical recordings (**g**) and representative frames from a P7 hemisection (**h**) showing a colour-coded BF activation pattern after stimulus delivery (top) and corresponding trace (bottom) at different time points. (**i**–**k**) Overview of the region of interest (ROI) arrangement: selection of an area comprising the HDB (horizontal diagonal band), VDB (vertical diagonal band) and medial septum (MS) regions of the BF [26] (**j**,**k**), segmentation of the designated region (**j**) and rasterisation to obtain ‘space-time’ maps of activity (**i**). The scale bar in (**h**) is 5 ms on the *x*-axis and 0.025 ΔF/F on the *y*-axis. The colour scale indicates depolarization (green to red). The white scale bar in (**i**,**j**) is 1 mm; colour bar units: ΔF/F_0_. LFP: local field potential.

**Figure 2 brainsci-08-00132-f002:**
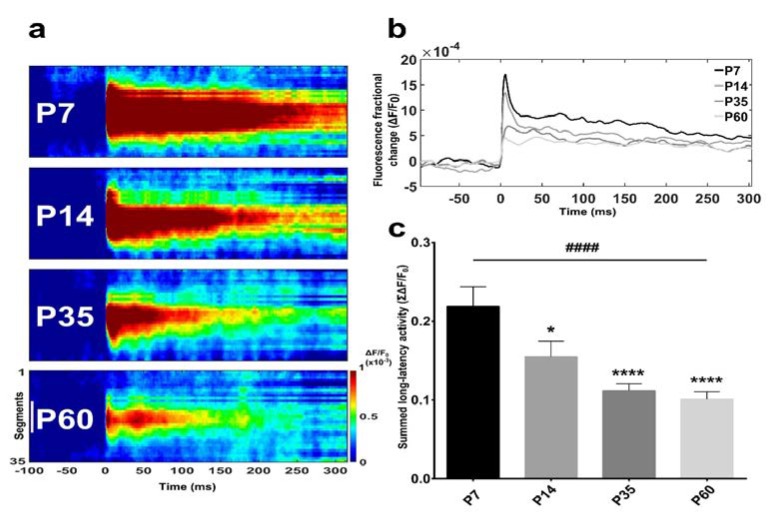
BF-evoked activity decrease is age-dependent. (**a**) Spatiotemporal analysis of BF-evoked response to stimulation. VSDI (voltage-sensitive dye imaging) data are shown as ‘space-time’ arrows of activity consisting of a bi-dimensional map defined by a hand-selected region of interest (see Figure 1j) over time (300 ms after the delivery of stimulus at T = 0 ms). The colour scale at the bottom indicates depolarizations as warm colours and hyperpolarisations as dark colours. The white scale bar at the bottom in (**a**) is 1 mm; colour bar units: ΔF/F_0_. Space-time maps relative to the four postnatal developmental stages show a reduction in assembly size. (**b**) Averaged response-time series from different age groups. Comparison of superimposed VSDI traces displaying a clear decrease in fluorescence fractional change from infant (P7) to young adult (P60) stage. (**c**) Bar graph indicating that the quantification of averaged summed fluorescence fractional change relative to P7, P14, P35 and P60 animals presented a significant decreasing trend between groups (trend analysis post-hoc test). Moreover, major variation were observed between P7 rats and older age-groups (Bonferroni’s post-hoc test). Asterisks indicate a difference between P7 and the other groups (Bonferroni’s post-hoc test), while hashes represent the linear trend among groups (trend analysis post-hoc test). Data are the mean ± SEM, *n* = (hemislices, rats): P7 = (30, 5); P14 = (31, 5); P35 = (33, 5); P60 = (30, 5). * *p* < 0.05, ****^,####^
*p* < 0.0001.

**Figure 3 brainsci-08-00132-f003:**
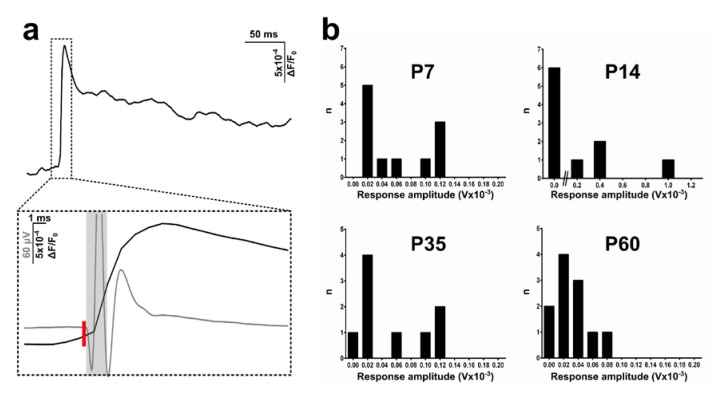
Comparison of VSDI (voltage-sensitive dye imaging) and electrophysiological signals and LFP (local field potential) recordings across different post-developmental stages. (**a**) Average of a representative VSDI trace from a P14 rat (**top**). The inset (**bottom**) shows VSDI and LPF traces superimposed and phase-locked from the same experiment. VSDI signal (black line) has a slower rising phase following stimulation (red bar), while field excitatory post synaptic potential (fEPSP) (grey line) reaches the peak before and has a faster decay. The light grey rectangle indicates a stimulation artefact. (**b**) Frequency distribution of post-stimulus response amplitudes at different ages. Early postnatal rats (P7–P14) are characterised by variable behaviour presenting a higher number of strong responses (>80 µV), while with maturation, this pattern shows a more conspicuous number of lower amplitude recordings. For (b): P7, *n* = 11 recordings; P14, *n* = 10; P35, *n* = 9; P60, *n* = 11.

**Figure 4 brainsci-08-00132-f004:**
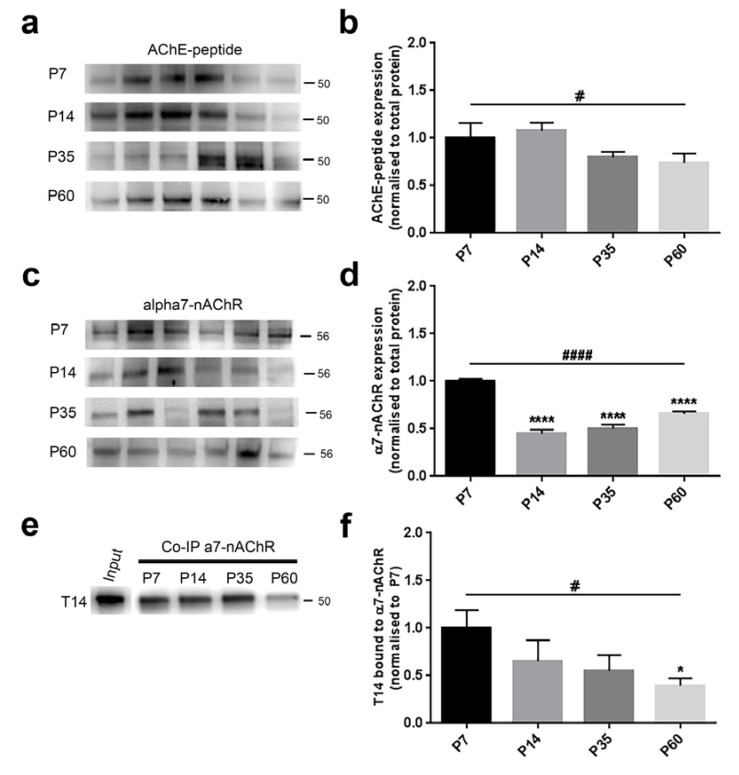
AChE-peptide and α7-nAChR levels and their interaction showed an age-dependent pattern during postnatal brain development. (**a**,**c**) Representative immunoblots showing the protein expression in six hemislices from one randomly-chosen animal per age. (**b**,**d**) Bar graphs indicating the protein profile related to four different ages. (**a**,**b**) AChE-peptide expression was not significantly affected across ages; however, its levels revealed a decreased tendency from early postnatal to young adult animals (trend analysis post-hoc test). (**c**,**d**) α7-nAChR levels were strongly decreased in an age-related manner (trend analysis post-hoc test). In particular, a significant difference was detected when comparing P7 rats to the other groups (Bonferroni’s post-hoc test). (**e**) Representative Co-IP blot showing the molecular interaction between T14 and the nicotinic receptor. (**f**) The complex T14/α7-nAChR showed a significant decreasing trend across ages (trend analysis post-hoc test) and also a clear change between the two extreme ages (Bonferroni’s post-hoc test). All values are normalised against the P7 group, considered as the control. Asterisks indicate a difference between P7 and the other groups (Bonferroni’s post-hoc test), while hashes represent the linear trend among groups (trend analysis post-hoc test). Data are indicated as the mean ± SEM. *n* = (hemislices, rats) is (24, 4). *^,#^
*p* < 0.05, ****^,####^
*p* < 0.0001.

**Figure 5 brainsci-08-00132-f005:**
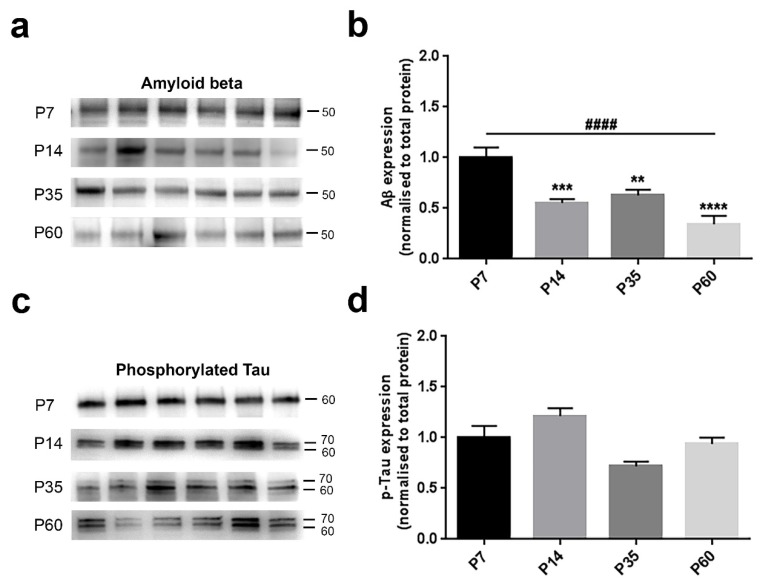
Amyloid beta and phosphorylated Tau levels are differently affected during postnatal brain development. (**a**,**c**) Representative immunoblots showing the protein expression in six hemislices from one randomly-chosen animal per age. (**b**–**d**) Bar graphs indicating the protein profile related to four different ages. (**a**,**b**) Amyloid levels were markedly influenced across ages, showing a substantial decreasing profile from P7–P60 (trend analysis post-hoc test) and also individual differences between P7 and older stages (Bonferroni’s post-hoc test). (**c**,**d**) Tau phosphorylation did not show any age-related trend, nor inter-individual variations. All values are normalised against the P7 group, considered as the control. Asterisks indicate a difference between P7 and the other groups (Bonferroni’s post-hoc test), while hashes represent the linear trend among groups (trend analysis post-hoc test). Data are indicated as the mean ± SEM. *n* = (hemislices, rats) is (24, 4). ** *p* < 0.01, *** *p* < 0.001 ****^,####^
*p* < 0.0001.

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
