# Peer review of "A Multidisciplinary Approach Reveals an Age-Dependent Expression of a Novel Bioactive Peptide, Already Involved in Neurodegeneration, in the Postnatal Rat Forebrain"

_brainsci, 2018, doi:10.3390/brainsci8070132_

Round 1

Reviewer 1 Report

This paper presents a study of the role of the endogenous T14 peptide in brain development, using voltage-sensitive dye imaging on rat brain slices containing the basal forebrain. The study was conducted with 44 rats at different postnatal ages. Experimental results suggest that T14 has an age-related effect in brain maturation.

The paper is well organized, but the readability is very poor, thus, I doubt its interest to general audience. 

Author Response

The paper is well organized, but the readability is very poor, thus, I doubt its interest to general audience. 

Answer: We recognize that this paper is a preliminary study, based on a new technical approach where we aimed to obtain a general overview about an age-related change in neuronal response and the T14 expression in brain slices containing the basal forebrain from four different age-groups. As such, we felt it particularly suitable for an edition focusing on mechanisms linked to AD, since we have long suggested that mechanisms of degeneration are similar to those of development, as described here. Given the narrow specificity of this topic we agree that the content  might not interest a general audience, but trust that this constraint does not detract from the validity of the data.  

Reviewer 2 Report

In the current study, the authors comprehensively investigated the role of T14 in the postnatal rat brain development. They found an age-dependent reduction of neuronal network activity which is paralleled by a decrease in the expression profile of the T14 in particular when bound to its target alpha-7 receptor. In conclusion, the results of this study suggest that T14 has an age-dependent expression and could be a critical signaling molecule during the brain development. 

The sample size is relatively small. There were only 4-5 animals per age-group for each experiment. 

Fig. 4 and 5: Please indicate why the protein expression of AChE-peptide, nAChR, Amyloid beta and Phosphorylated Tau in six hemislices from the same animal shows a significant difference. 

In this study, the authors investigate the expression of AD-related proteins. It seems like that aging animal could be a better model for the current study. It will be interesting to compare the expression profile of T14 between young and much older rats.

Please explain how the age-dependent reduction of the neuronal network was associated with the decrease of T14 expression during the brain development base on the results of the current study.

What is the relationship between p-Tau and T14? 

Author Response

Comments and Suggestions for Authors

In the current study, the authors comprehensively investigated the role of T14 in the postnatal rat brain development. They found an age-dependent reduction of neuronal network activity which is paralleled by a decrease in the expression profile of the T14 in particular when bound to its target alpha-7 receptor. In conclusion, the results of this study suggest that T14 has an age-dependent expression and could be a critical signalling molecule during the brain development. 

The sample size is relatively small. There were only 4-5 animals per age-group for each experiment.

Answer: We agree with the reviewer’s comment that the sample size is relatively small. However, this was due to the complex nature of the experimental approach involving two very different and time-consuming techniques in the same tissue. Therefore, we first aimed to investigate preliminarily this combination of methods before planning a more complete study.

Fig. 4 and 5: Please indicate why the protein expression of AChE-peptide, nAChR, Amyloid beta and Phosphorylated Tau in six hemislices from the same animal shows a significant difference.

Answer: The difference in protein expression observed in the six hemislices can be related to the fact that the 6 hemislices per animal for both VSDI and WB techniques were randomly selected for this study. Given the quasi-physiological nature of the preparation and complexity of procedure, a degree of variability in a range of parameters would be expected. In particular, anatomical changes between different hemislices could explain the fluctuations in protein expression seen in Fig 4 and 5; for this reason we decided to average protein levels of each hemisection for each group age. 

In this study, the authors investigate the expression of AD-related proteins. It seems like that aging animal could be a better model for the current study. It will be interesting to compare the expression profile of T14 between young and much older rats.

Answer: We acknowledge that the investigation on AD-related proteins in AD-like models would provide a more complete overview of the T14 pathway. However, ageing is not an established model of AD, nor is AD a natural consequence of ageing, - rather a disease of old age. As such we would need to work with a specific AD model, for which there is as yet no consensus.  The most obvious start for this study on brain development was with wildtype animals, grouped within a small age-window: our first aim was to see if any change was detectable in AD-related proteins that are also involved in physiological processes, such as those in brain development, which have previously been linked to AD. We agree with the reviewer that other studies using more extreme ages (including wildtype and AD-related animal models) could further characterize T14 expression levels and its possible involvement in brain maturation events.

Please explain how the age-dependent reduction of the neuronal network was associated with the decrease of T14 expression during the brain development base on the results of the current study.

Answer: The observed reduction in neuronal activity, from P7 to P60 animals, paralleled the decreasing T14 levels in the investigated time window. In support of the T14 effect in modulating network response we have previously shown that application of exogenous T30 reduces the activity of neuronal assemblies, while a cyclic compound derived from T14 (NBP14) induces a recovery (Badin et al., 2016).

What is the relationship between p-Tau and T14?

Answer: We have established in various publications and preparations that T14 enhances calcium influx: as such it would trigger, inter alia, the hyperphosphorylation of tau (Garcia-Rates et al, 2016) There is no direct attempt here to explore this sequence of events any further. However we have previously described how T30 modulates GSK3 expression, one of the kinases implied in Tau phosphorylation. For these reasons we explored also in this study p-Tau expression and its variation in different ages and suggested that endogenous T14 could contribute to the modulation of p-Tau levels.